# The Effect of Alternative Splicing Sites on Mirtron Formation and Arm Selection of Precursor microRNAs

**DOI:** 10.3390/ijms25147643

**Published:** 2024-07-12

**Authors:** Luca Gál, Anita Schamberger, Gerda Wachtl, Tamás I. Orbán

**Affiliations:** 1Gene Regulation Research Group, Institute of Molecular Life Sciences, HUN-REN Research Centre for Natural Sciences, 1117 Budapest, Hungary; 2Doctoral School of Biology, Institute of Biology, ELTE Eötvös Loránd University, 1117 Budapest, Hungary; 3Institute of Genetics and Biotechnology, Hungarian University of Agriculture and Life Sciences, 2100 Gödöllő, Hungary

**Keywords:** mir-877, mir-33b, RNA interference, RNAi, miRNA, microRNA, mirtron, Drosha

## Abstract

Mirtrons represent a subclass of microRNAs (miRNAs) that rely on the splicing machinery for their maturation. However, the molecular details of this Drosha-independent processing are still not fully understood; as an example, the Microprocessor complex cannot process the mirtronic pre-miRNA from the transcript even if splice site mutations are present. To investigate the influence of alternative splicing sites on mirtron formation, we generated Enhanced Green Fluorescent Protein (EGFP) reporters containing artificial introns to compare the processing of canonical miRNAs and mirtrons. Although mutations of both splice sites generated a complex pattern of alternative transcripts, mirtron formation was always severely affected as opposed to the normal processing of the canonical hsa-mir-33b miRNA. However, we also detected that while its formation was also hindered, the mirtron-derived hsa-mir-877-3p miRNA was less affected by certain mutations than the hsa-mir-877-5p species. By knocking down Drosha, we showed that this phenomenon is not dependent on Microprocessor activity but rather points toward the potential stability difference between the miRNAs from the different arms. Our results indicate that when the major splice sites are mutated, mirtron formation cannot be rescued by nearby alternative splice sites, and stability differences between 5p and 3p species should also be considered for functional studies of mirtrons.

## 1. Introduction

MicroRNAs (miRNAs) are short; on average, they are 19–24 nucleotide (nt)-long single-stranded RNA molecules. They form ribonucleoprotein complexes with members of the Argonaute protein family, and they initiate gene silencing predominantly at the post-transcriptional level [1,2,3]. MiRNAs belong to a distinct pathway of RNA interference, which has evolved to control endogenous genes and forms an intricate regulatory network, the complexity of which is comparable to that of transcription factors [4,5]. The primary-miRNA (pri-miRNA) transcripts are generated from endogenous loci, and their maturation occurs via two consecutive endonucleolytic cleavage steps. In animals, the first one is carried out in the nucleus by the RNase III enzyme Drosha, associated with the DiGeorge Syndrome Critical Region 8 (DGCR8) protein in the Microprocessor complex, resulting in the formation of a hairpin-structured precursor-miRNA (pre-miRNA). The second cleavage occurs in the cytoplasm by another RNase III enzyme Dicer into a 19–24 nucleotide-long miRNA duplex structure [6], followed by the association with an Argonaute protein. After strand separation and selection, the mature miRNA becomes part of the RNA-induced silencing complex (RISC). By scanning RNA molecules in the cytoplasm, the RISC finds its targets via sequence complementarity between its miRNA and a 3′ untranslated region (UTR) motif of a protein-coding gene. However, miRNA-target interactions can vary, enhancing the diversity in the functionality of miRNAs [7]. As a consequence of these interactions, translation is blocked, or the degradation of the mRNA target is initiated [8,9]. The efficiency of silencing depends on the number of miRNA sites in the 3′ UTR (targeted by the same or different miRNA species), and such complex interactions lead to an elaborate fine-tuning of mRNA expressions in the cells [10,11,12].

The above-outlined enzymatic processes represent the “canonical” maturation pathway of miRNAs. However, miRNAs could also be formed in alternative routes, which usually means the substitution of one of the two cleavage steps by other mechanisms, thereby representing Drosha- or Dicer-independent miRNA formations [13,14]. The most prominent Drosha-independent route is the mirtron pathway, where the pre-miRNA sequence itself has become an intron of a host gene, and the Microprocessor cleavage is replaced by the splicing machinery. Mirtrons were discovered in *Drosophila melanogaster* and in *Caenorhabditis elegans* [15,16,17] but were later predicted to be present also in species having generally longer introns, including vertebrates and plants [18,19,20,21,22,23]. Later studies provided functional evidence for mirtron formation in mammalian species, and they all agreed that bioinformatics predictions are insufficient to prove the presence of functional mirtrons [24,25,26,27]. Further studies showed that besides “conventional” mirtrons, other types called “tailed” mirtrons also exist; in those cases, additional sequences are also present between the mirtronic pre-miRNA and either the 5′ or the 3′ splice sites. Although they still require splicing for processing, additional trimming of the 5′ and/or the 3′ tail(s) is necessary to form the fully matured pre-miRNA [28,29,30,31,32,33]. Mirtrons are also considered promising candidates in gene therapy as they can be a source of miRNAs released from the introns of other therapeutic target proteins [30,34,35]. However, it is crucial to understand their structural motifs and the details of their maturation steps in order to avoid the formation of improper small RNA species, thereby causing off-target effects, a relevant problem frequently hindering miRNA-based targeting [36]. 

The structural details and constraints of mirtron processing and its strict dependence on splicing are still not fully understood, especially in light of the fact that intronic miRNAs represent the predominant types among intragenic miRNAs [37]. It was shown earlier that for long introns containing pre-miRNAs, the Microprocessor complex can remove the precursor miRNAs from the transcript even if splicing is inhibited [38]. In the case of mirtrons, however, mutations in the flanking splice sites still render the mirtronic pre-miRNA inaccessible for Drosha cleavage [39]. The question arises whether it is caused by a steric hindrance of a strong secondary structure, by the short hairpin structure that is suboptimal for Drosha processing, or rather by a strongly associated RNA binding protein (such as a still present splicing factor) that could prevent Microprocessor activity on a mirtronic locus. Another important but unexamined aspect is the mirtronic “behavior” connected to alternative splicing; if nearby alternative splice sites exist, they may initiate tailed mirtron formation, which could rescue the mirtron processing pathway when the major splice sites are mutated. With 5′ tailed mirtrons predicted to be more abundant than even conventional mirtrons [40], this issue merits further investigations. To address this problem, we used *EGFP*-based reporter assays to investigate the effects of alternative splice sites on canonical and mirtronic miRNA formation in the absence of the major cis-acting splicing elements. We inserted different artificial introns into the *EGFP* coding sequence and generated 5′ and 3′ splice site (SS) mutations (individually or in combination) to measure miRNA expression from such loci. In many cases, the mutations resulted in the appearance of various alternative splice products that had no negative effect on canonical miRNA formation from a longer intron. In the case of a mirtronic intron, however, disrupting the major splice sites abolished miRNA formation from both pre-miRNA arms, although to a different extent. Our results support the model where the major splice site mutations at a mirtron locus strictly prevent miRNA formation, even if weak alternative splice sites are used in the nearby flanking regions.

## 2. Results

### 2.1. Mutations in the 5′ or 3′ Splice Sites Initiate the Usage of Alternative Donor or Acceptor Sites

To investigate the effect of splice site mutations, we generated *EGFP*-based reporters containing short introns with a size comparable to that of mirtrons (<500 bp) and transfected them into HEK-293H cells. We used the sequence of the hsa-mir-877 mirtron, the short sixteenth intron of the *Sterol regulatory element-binding protein 1* (*SREBF1*) gene containing the hsa-mir-33b (as a canonical miRNA control, see [25]), and the sixth intron of the *NADH:ubiquinone oxidoreductase core subunit F1* (*NDUFS1*) gene (*NADi6*, as a natural short intron control, see [26]). In these constructs, the disturbance of splicing can be monitored by detecting the presence or absence of EGFP signals via fluorescence microscopy. In all cases, single point mutations of the 5′ or 3′ splice sites resulted in the lack of green fluorescence, indicating the mutation or the absence of the EGFP protein (Figure 1). Subsequently, using primers targeting the two artificial exonic sequences of *EGFP*, we analyzed the mRNA splicing patterns from the constructs. For the wild-type (normal) reporters, we could detect very efficient splicing for the *NADi6* and the hsa-mir-33b introns, whereas the splicing reaction was moderately efficient for the hsa-mir-877 mirtron, as a significant amount of the unspliced product could also be seen during gel electrophoresis (Figure 2). For the mutant constructs, however, we detected different splice products depending on the intronic sequence and also on the position (5′ or 3′ SS) of the mutation. For the *NADi6* intron, the 5′ SS mutation completely abolished splicing, whereas, for the mirtron and the hsa-miR-33b-containing intron, a single alternative splice product could be detected (Figure 2), as a consequence of an alternative 5′ SS located upstream of the canonical one (Appendix A). The usage of the alternative 5′ SS, in the absence of the canonical 5′ SS, resulted in an in-frame deletion in the coding sequence of EGFP. This led to a non-functional protein, as indicated by the lack of fluorescence signal for this construct using microscopy (Figure 1).

Introducing a single point mutation to disrupt the identified alternative 5′ SS generated a valine to leucine missense mutation (due to sequence constraints). However, it still led to a functional EGFP and the normal splice product because the splicing machinery could utilize the canonical 5′ SS. Mutating both the canonical and the alternative 5′ SSs, however, resulted in a mixture of very weak (if any) alternative splice products, even for the miRNA-containing introns (Figure 2).

Concerning the 3′ SS mutation, the choice of alternative acceptor sites seemed more random, showing substantial variations among experiments, but could occasionally result in discrete alternative products (Figure 2 and Appendix A). Mutating all potential 5′ and 3′ splice sites prevented the formation of splicing products in the case of the *NADi6* construct, whereas for the miRNA-containing introns, it still led to the formation of faint alternative splice products, which could be considered as “stochastic background splicing noise” (not all products could be unambiguously identified due to the limitations in detection methodology) (Figure 2). Although it clearly abolished the formation of a functional protein product, it raised the question of whether such a complex splicing pattern could still contribute to the formation of functional miRNAs, especially for the splicing-dependent mirtron.

### 2.2. Interactions between the Splicing Apparatus and the miRNA Maturation Pathway

To test the miRNA formation from the constructs carrying different splice site mutations, we measured the level of mature small RNA species generated from both arms of the pre-miRNA loci by quantitative real-time PCR (qRT-PCR). The reporter constructs contained an *mCherry* expression cassette with a canonical hsa-miR-512 generated from its 3′ UTR in order to control for transfection efficiency, as well as to normalize for mRNA/miRNA expression from the vector (Figure 1A). In the case of the canonical miRNA (hsa-mir-33b), both the dominant guide (5p) and the passenger (3p) strands could be detected in the cells, and the splicing mutations did not lower their expression levels. In fact, their steady-state levels showed a tendency to increase from the mutant constructs, although the changes were not significantly different in most cases (Figure 3A,B). Moreover, the expression levels did not correlate with the steady-state levels of the host (EGFP) mRNA, as even in the case of the canonical 5′ SS mutant where the EGFP mRNA expression was significantly lower, the formation of the hsa-mir-33b species was not disturbed (see Appendix A, Figure 3A,B).

For the hsa-mir-877 mirtron, however, splice mutations strongly impaired miRNA formation. It was most noticeable for the dominant guide species (also formed here from the 5p arm), where mutating even the alternative 5′ SS could significantly lower its steady-state level (Figure 3C). The level of the miRNA generated from the passenger (3p) strand showed a strong decrease in the 3′ SS mutants; however, contrary to expectations, it was much less sensitive to mutations in the 5′ SSs (Figure 3D). This strand bias was not expected from a mirtronic locus and prompted us to test whether the formation of both 5p and 3p species occurs via a Drosha-independent pathway.

### 2.3. Arm Selection of Mirtron-Originated miRNAs Is Determined via Drosha-Independent Mechanisms

The crucial difference between the canonical and mirtronic pre-miRNA formation is the dependence on the Microprocessor complex and, most importantly, on its central component, the Drosha endonuclease. To investigate this factor in our system, we generated siRNA knockdown of Drosha in the cells and tested miRNA formation from our splicing reporters. The efficiency of the knockdown was indicated by the significant reduction of Drosha mRNA levels (Figure 4A) and functionally proven by the significantly lowered canonical miRNA species, the hsa-miR-512 control from our vector constructs (Figure 4B). On the other hand, neither the 5p nor the 3p arm species of the hsa-miR-877 mirtron showed a decreased expression in the Drosha knockdown experiments, and their levels were still solely determined by the active or mutated statuses of the canonical splice sites (Figure 4C,D). These results suggested that the alternative 5′ SS did not initiate the formation of an efficient 5′-tailed mirtron, and the biased effect detected on the mirtron-derived 5p or 3p miRNA species were the results of Drosha-independent mechanism(s).

## 3. Discussion

Since its discovery, several aspects of the mirtron pathway have been deciphered, but the molecular details of its splicing dependence remain incompletely understood [40]. An earlier study showed a strong association between the Microprocessor complex and the spliceosome, suggesting that for pre-miRNAs in long introns, Drosha-mediated cropping precedes splicing [39]. However, efficient recognition of the stem-loop structure by Drosha is not optimal for mirtrons, as, in general, they are too short to form the necessary length of helical turns [15,41]. The splicing machinery and miRNA processing often target the same transcripts, sometimes overriding each other [42]. This complexity is heightened when alternative splicing produces different isoforms from the same locus, each containing different pre-miRNAs. This regulation is crucial for mirtrons, whose formation is determined by splicing; shifts in splice site usage can significantly hinder their miRNA production [43].

In this study, we investigated the role of alternative splice sites in forming mirtronic pre-miRNAs, particularly when the major splice donor or acceptor sites are mutated. We hypothesized that if a nearby alternative splice site is located close to the major ones, the short additional sequence could lead to a tailed-mirtron formation, allowing pre-miRNA processing from the locus. To test this, we generated several *EGFP*-based reporters with either a mirtron, a canonical miRNA-containing intron, or a short “miRNA-less” intron. The first important observation was that regardless of the same exonic sequences, reporters with a pre-miRNA in the intron generated more alternative splice sites than those with canonical short introns. Notably, an alternative 5′ SS in the upstream exonic sequence was detected for the hsa-mir-33b and hsa-mir-877 reporters but not for the *NADi6* intron reporter (Figure 2). The splicing efficiency of this alternative donor site correlated with the original 5′ SS efficiency; it was nearly 100% for the hsa-mir-33b reporter but much weaker for the hsa-mir-877 mirtron. This suggests that different intronic splicing enhancer or silencer elements may contribute to this phenomenon [44]; however, the prediction of such elements is still far from reliable [45], awaiting further systematic study in this regard.

Another unusual aspect of the alternative 5′ SS is that it is exclusively used only when the original splice site of the pre-miRNA-containing intron is mutated. This is unexpected, as splice site definition predominantly occurs co-transcriptionally, suggesting that a favorable upstream splice site would be used in some transcripts [46]. However, we did not detect any alternative transcripts in normal constructs, indicating that the major splice donor site dominates over the upstream signal. One explanation could be that the strong secondary structure of the pre-miRNA or an associated splicing factor suppresses the upstream splice donor site. Complex regulations for splice-site-overlapping pre-miRNA hairpins (SO-miRNAs) could lead to direct *cis* competition with the splicing machinery [42,47]. For example, Serine/Arginine-Rich Splicing Factor 1 (SRSF1) regulates the inclusion of exons containing SO-miRNA miR-222, thereby suppressing the expression of miR-222 [48]. In addition, a recent study showed a similar regulation by a 3′-tailed mirtron in *Drosophila*, where miR-1017 binds to its own 5′ SS, inhibiting splicing and preventing its own formation [49]. Nevertheless, such an elaborate mechanism for a conventional mirtron like hsa-miR-877 would require further experimental validation.

Regarding the effect of splicing on intronic miRNA processing, our findings align with previous data: when pre-miRNA sequences are unaffected by the mutations, canonical miRNAs are efficiently processed from introns, irrespective of splice patterns and mRNA isoforms. However, severe splicing disruptions tend to increase hsa-miR-33b expression (Figure 3A,B), suggesting interference with processing by the Microprocessor complex [42]. Our analysis of the hsa-mir-877-containing reporter supports prior observations [25,26,27] that this species functions as a bona fide mirtron; mutations in splice sites strongly inhibit miRNA formation (Figure 3C,D), independent of the presence of functional Drosha (Figure 4). Nevertheless, mutations sometimes lead to alternative or cryptic splice site usage nearby, potentially forming a tailed-mirtron, which would allow pre-miRNA processing. Notably, a canonical 5′ splice site mutation hypothetically added a 45-nucleotide extension between the intron’s 5′ end and the pre-miRNA (Appendix A), suggesting a 5′-tailed mirtron formation. However, mature miRNA species were not detected, indicating that this alternative intron cannot enter the miRNA maturation pathway. Despite genome-wide suggestions that 5′-tailed mirtrons outnumber conventional mirtrons, their structural features and the nucleases responsible for their processing remain poorly characterized [40]. Therefore, further investigations are crucial to understand why our alternative intron does not function as a 5′-tailed mirtron. 

Another significant finding was that even when all potential splice sites were mutated, miRNA still could not be formed from the mirtronic locus. It could be assumed that in such cases, a long mRNA containing a pre-miRNA hairpin could still be processed by Drosha via the canonical pathway; however, this was not the case for the hsa-mir-877 locus. Possible explanations could be that the pre-miRNA sequence is still too short for effective Drosha processing [41], or the processing can be blocked in a later step in the pathway. Previous studies have shown that the presence of an endogenous N-terminal truncated Dicer isoform elevated the level of mirtronic miRNAs over canonical ones during mouse development, indicating that the canonical Dicer isoform is not suitable for recognizing mirtron hairpin structures [50,51]. Alternatively, essential elements like the branch point and the polypyrimidine tract (all required for splicing [46]) remain associated with splicing factors that would inhibit Drosha activity. It could also be argued that the low and stochastic splicing activity detected might interfere with the Microprocessor complex, although the majority of mRNAs being unprocessed makes this less likely. Another puzzling aspect of our study was the atypical results observed with the mirtronic hsa-miR-877-3p arm; mutations in the 5′ donor sites led to a decrease in its expression, though not as drastic as with the 3′ SS mutation (or the 5′ splice site mutation for the hsa-miR-877-5p arm). Since these effects were also observed in knockdown experiments, they are likely caused by Drosha-independent mechanisms. One potential explanation could involve selective processing of the 5p and 3p arms, where factors associated with the 5p or upstream sequences may be more sensitive to mutations in the 5′ SS, while 3p processing may be less affected, resulting in seemingly higher steady-state levels of the 3p species. However, without experimental confirmation, this remains speculative. Overall, our study underscores the limited understanding of the structural constraints of mirtrons. High-throughput studies have provided comprehensive data on the structural features of miRNAs [52], but specific knowledge regarding mirtrons, which constitute a large subclass of miRNAs, remains incomplete. Our findings indicate that the splicing dependence of mirtron processing is stricter than previously thought, as nearby alternative splice sites cannot compensate for pre-miRNA removal when major splice sites are mutated. Additionally, since artificially designed mirtrons are considered tools for molecular targeting and therapeutic purposes [30,34], future research on mirtron maturation and function should comprehensively investigate both arms of the pre-miRNA, as strand selection and differences in stability could significantly impact study outcomes.

In conclusion, we have shown that mutations in the canonical 5′ and 3′ splice sites lead to the use of alternative splice sites, resulting in a more complex splicing pattern for all three types of introns. We found that miRNA maturation from the canonical hsa-miR-33b miRNA-containing intron was tendentiously increased in the case of splicing deficiency, indicating competition between the splicing machinery and the Drosha/DGCR8 Microprocessor complex during transcription. Additionally, we observed that the 5′ SS mutation affected miRNA production from the 3p arm of the hsa-miR-877 mirtron locus less; however, this phenomenon was not due to either Drosha dependency or the generation of a 5′-tailed mirtron via alternative 5′ SS usage. These findings are consistent with the current knowledge about mirtron generation and its strict splicing dependence, but they also highlight the importance of considering arm selection regulation during functional studies.

## 4. Materials and Methods

### 4.1. Plasmid Constructs

The *Sleeping Beauty* (SB) transposon-based expression vectors contain the CMV promoter-driven *mCherry* gene that carries the precursor sequence of the placenta-specific hsa-mir-512 miRNA in its 3′ UTR. mCherry and hsa-miR-512-3p were used as internal controls in the subsequent experiments. Downstream of this transcription unit, the vectors also contain another CAG promoter-driven expression cassette, where the originally intronless *EGFP* gene is divided into two artificial exons by introducing a PvuII restriction site [24,25], allowing cloning a mirtron, an miRNA-containing intron, or a short intron sequence between the two artificial exons. The sequences of human mirtron hsa-mir-877 and the intron containing the hsa-mir-33b canonical miRNA were amplified from plasmids that were used in our previous studies [24,25]. In addition, the short 6th intron of human *NDUSF1* intron 6 (NADi6) was amplified from the pEGFP-NAD plasmid (Addgene ID: 58345; Addgene, Watertown, MA, USA) and inserted into the EGFP reporter as a short intron control without an miRNA precursor. 

To generate the splice site mutants of the different intron-containing *EGFP*, we combined the mutation of the canonical 5′ splicing site (GT > TG), the alternative 5′ splice site (C > G), and the 3′ splice site (GA > AG) (Figure 1A). The normal construct does not contain any of the mutations. The 5′ SS mutant contains the mutation of the canonical 5′ SS, the alternative 5′ SS (alt. 5′ SS) mutant contains the mutation of the alternative 5′ SS site, whereas the double 5′ SS mutant contains both. The 3′ SS mutant contains the mutation of 3′ SS, whereas the total mutant contains the two 5′ SS mutations and the 3′ SS mutation (summarized in Figure 1A). All expression vectors used in this study were created by the NEBuilder^®^ HiFi DNA Assembly method (New England Biolabs, Ipswich, MA, USA), and the alternative 5′ SS site was introduced by site-directed mutagenesis. Vector sequences were verified by Sanger sequencing (Microsynth AG, Wien, Austria). Primers used for the assemblies are listed in the Appendix A.

### 4.2. Cell Culturing and Treatments

The HEK-293H (primary embryonic human kidney) cell line was cultured in Dulbecco’s modified Eagle’s medium (DMEM) supplemented with 10% fetal bovine serum (FBS), 1% L-glutamine, and 1% penicillin–streptomycin (Thermo Fisher Scientific, Waltham, MA, USA). For plasmid transfections, 5 × 10^5^ cells were seeded onto 6-well plates in 2 mL of antibiotic-free DMEM culture medium containing FBS. The next day, 1.2 µg of plasmid was transfected into the cells using the Lipofectamine™ LTX Reagent with PLUS™ Reagent, following the manufacturer’s instructions (Thermo Fisher Scientific, Waltham, MA, USA). At 48 h post-transfection, EGFP and mCherry fluorescence were detected by a IX51 fluorescence microscope (Olympus Corporation, Shinjuku, Tokyo, Japan), and cells were subjected to further experiments or analyses.

For knockdown (KD) experiments, 2.5 × 10^5^ cells were seeded onto 12-well plates in 1 mL of antibiotic-free DMEM culture medium containing FBS. The next day, 50 nM of siRNA-targeting Drosha (cat. #4390824) and negative control (cat. #4390843) were transfected into the cells using Lipofectamine™ RNAiMAX Transfection Reagent, following the manufacturer’s instructions (Thermo Fisher Scientific, Waltham, MA, USA). At 24 h post-transfection, the KD cells were reseeded onto 6-well plates and transfected with the splicing plasmids, as described above.

### 4.3. RNA Extraction

Total RNA was isolated from cultured cells using TRIzol™ RNA Isolation Reagents (Thermo Fisher Scientific, Waltham, MA, USA) following the manufacturer’s instructions. RNA integrity was analyzed by agarose gel electrophoresis, and concentration and sample purity were measured by a Nanodrop spectrophotometer (Thermo Fisher Scientific, Waltham, MA, USA).

### 4.4. Reverse Transcription

To remove genomic DNA contaminations, total RNA samples were treated with DNase I (New England Biolabs, Ipswich, MA, USA) at 37 °C for 1 h. For mRNA analysis, 1 μg total RNA was reverse transcribed by random oligomers using the High-Capacity cDNA Reverse Transcription Kit (Thermo Fisher Scientific, Waltham, MA, USA). For miRNA analysis, 10 ng of total RNA was applied for cDNA synthesis by TaqMan™ Advanced miRNA cDNA Synthesis Kit (Thermo Fisher Scientific, Waltham, MA, USA). cDNA samples were diluted 1:10 before subsequent amplifications.

### 4.5. Splicing Detection Assay

Using primers (Sigma-Aldrich, St. Louis, Missouri, USA) targeting the two artificial exonic sequences of EGFP (Figure 2A), we amplified the entire or the differently spliced EGFP-intron/mirtron cassette by PCR (EGFP-sequencing-For: 5′-AAGGGCGAGGAGCTGTTCA-3′, EGFP-sequencing-Rev: 5′-TCCATGCCGAGAGTGATCC-3′). We used cDNA to detect mRNA splicing isoforms and to estimate splicing efficiency; we also used plasmid DNA templates as non-splicing controls. Amplified splicing products were visualized using agarose gel electrophoresis.

### 4.6. Quantitative Real-Time PCR (qPCR)

For mRNA expression analysis, qPCR was carried out using TaqMan^®^ Gene Expression Master Mix (Thermo Fisher Scientific, Waltham, MA, USA) according to the manufacturer’s instructions. Pre-designed TaqMan^®^ assays were used in the case of Drosha (Assay ID Hs00203008_m1), mCherry (Assay ID: Mr07319439), and POLR2A (Assay ID: Hs00172187_m1), while a previously designed custom-made TaqMan assay was used in the case of EGFP [53]. For miRNA quantification, expression analysis was performed by TaqMan™ Fast Advanced Master Mix for qPCR (Thermo Fisher Scientific, Waltham, MA, USA). TaqMan^®^ Advanced miRNA Assays were used in the case of hsa-miR-33b-5p (Assay ID: 478479_mir), hsa-miR-33b-3p (Assay ID: 478832_mir), hsa-miR-877-5p (assay ID: 478206_mir), hsa-miR-877-3p (assay ID: 477856_mir), and hsa-miR-512-3p (Assay ID: 478971_mir), while the U6 snRNA TaqMan™ microRNA Control Assay was applied as an endogenous control (Assay ID 001973). qPCR measurements were run on the QuantStudio™ 3 platform (Thermo Fisher Scientific, Waltham, MA, USA) according to the manufacturer’s instructions.

In our qPCR measurements, we used the ΔΔCt method for quantification [54]. For miRNA expression analysis, we used the hsa-miR-512-3p as a control, and in Drosha KD experiments, U6 served as the control. For quantifying EGFP expression, mCherry was the control gene, whereas for measuring Drosha expression, POLR2A served as the control gene. We normalized the mRNA and miRNA expression levels to the expression level of the control sample (normal) separately in every measurement. For statistical analysis, we used a one-sample t-test to compare the miRNA or mRNA expressions of the mutant constructs to that of the normal construct, which was considered to be 1 in every measurement. When comparing expression changes among different mutant groups affected by Drosha KD, we used Student’s *t*-test.

## Figures and Tables

**Figure 1 ijms-25-07643-f001:**
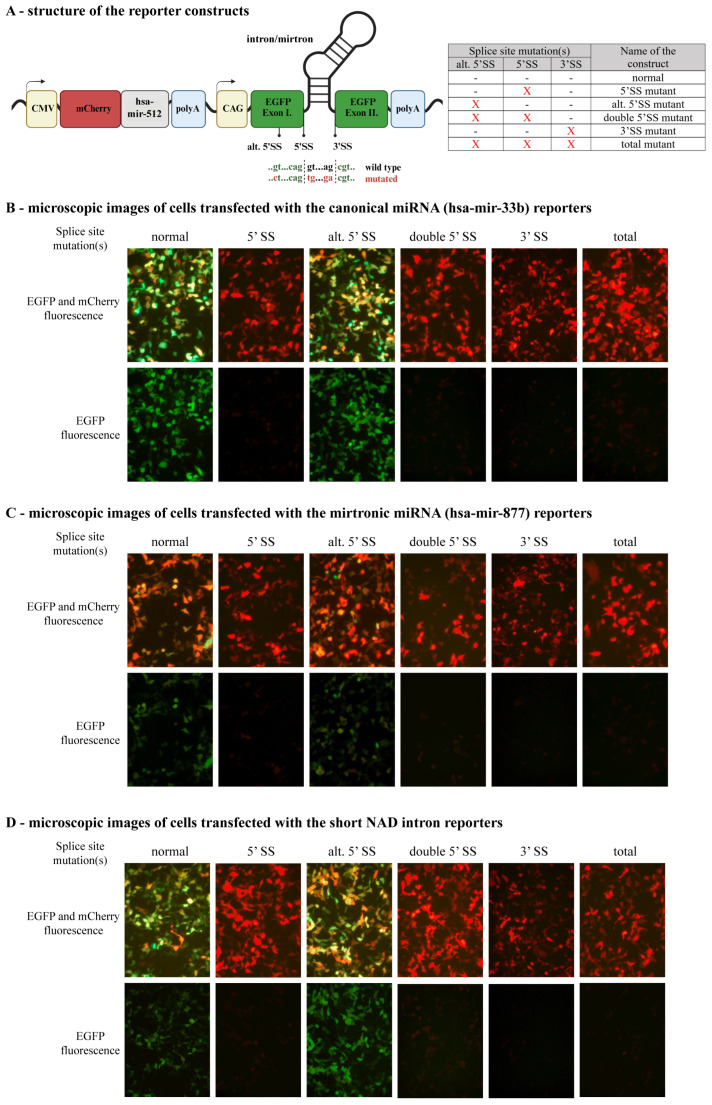
Testing the functionality of the various *EGFP* reporter constructs. (**A**) Schematic structure of the reporter vectors. To normalize for transfection efficiency and expression levels, all reporters contain an *mCherry* expression cassette with the canonical hsa-mir-512 miRNA processed from the 3′ UTR. For the *EGFP* reporters containing an artificial intron, the positions of the different splice site mutations are depicted on the left diagram, and their presence in various combinations is summarized in a table format on the right. (**B**) Detecting the expression of functional EGFP by fluorescence microscopy in the case of the hsa-mir-33b canonical miRNA expressing reporters carrying different splice site mutations (indicated on the top). Upper images show both red and green fluorescence, whereas the lower ones show only the green fluorescence generated from the EGFP protein. (**C**) Detecting the presence of functional EGFP for the hsa-mir-877 mirtronic miRNA-containing reporters. (**D**) Detecting EGFP signals for the short miRNA-less *NADi6* intron-containing reporters. All images were taken at 40× magnification.

**Figure 2 ijms-25-07643-f002:**
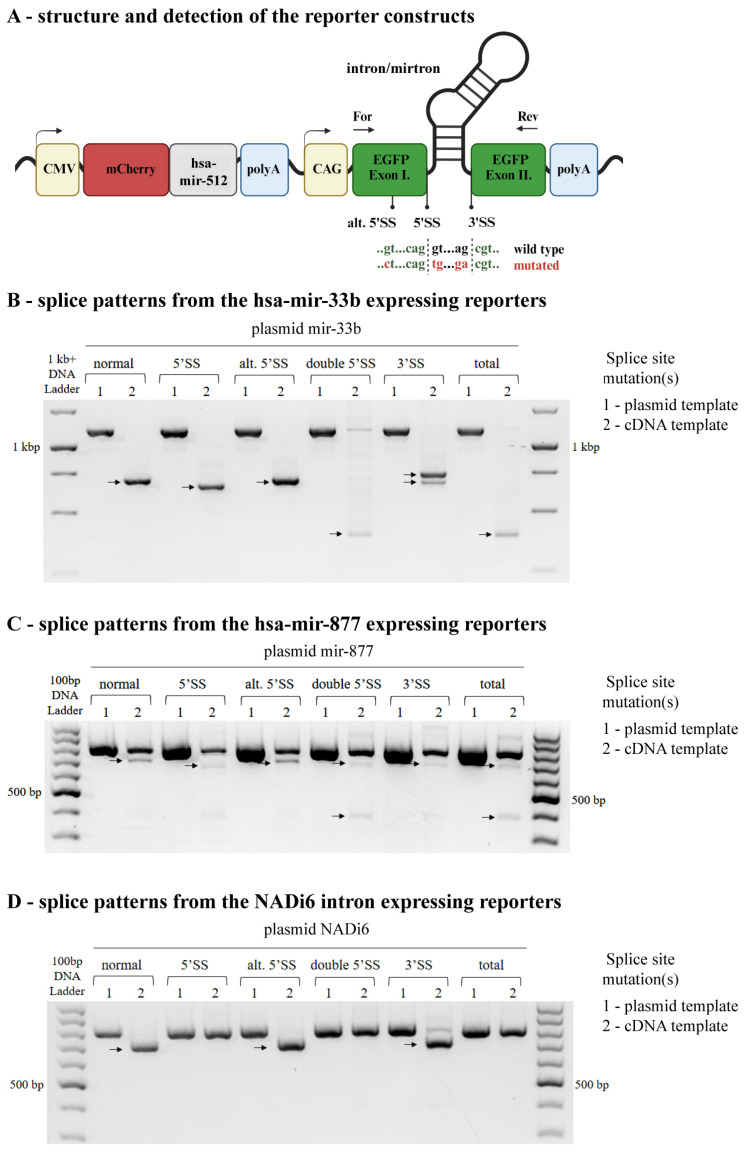
Investigating the splice patterns expressed from the various reporters. (**A**) Exonic primers used to amplify the splicing variants are depicted on the schematic diagram of the reporters (“For” and “Rev”). PCR products run on agarose gels to detect the different mRNA variants formed from the (**B**) hsa-mir-33b-, the (**C**) hsa-mir-877-, or the (**D**) *NADi6*-containing reporters. As a control, amplification was also carried out from the plasmid template, resulting in a product with the same size as the unspliced mRNA. Arrows point to discrete splice products, most of which could be identified by Sanger sequencing (see Appendix A).

**Figure 3 ijms-25-07643-f003:**
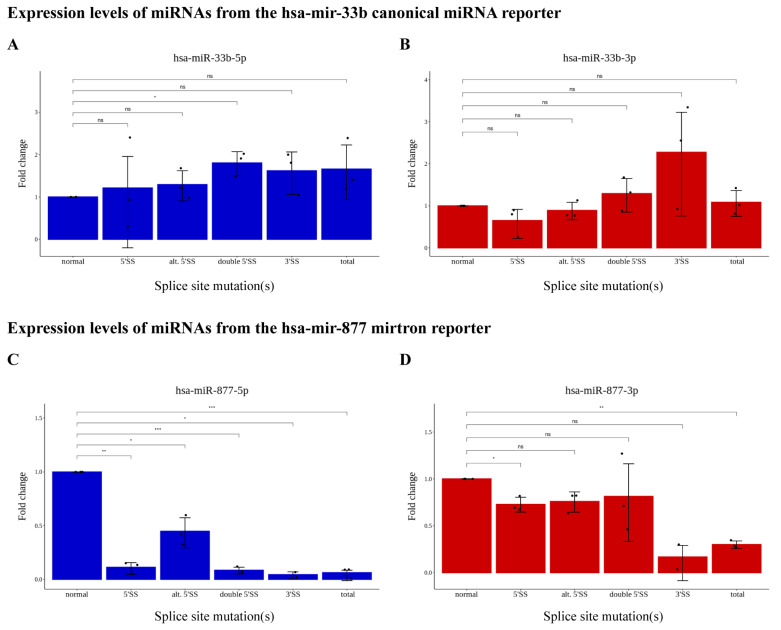
Relative expression levels of the (**A**) hsa-miR-33b-5p and the (**B**) hsa-miR-33b-3p, or the (**C**) hsa-miR-877-5p and the (**D**) hsa-miR-877-3p miRNA species expressed from the corresponding reporter constructs. Mean values of three independent biological experiments are shown, the black dots indicate the results of individual experiments; error bars represent the 95% confidence intervals. ns: not significant; *: *p* < 0.05; **: *p* < 0.01; ***: *p* < 0.001.

**Figure 4 ijms-25-07643-f004:**
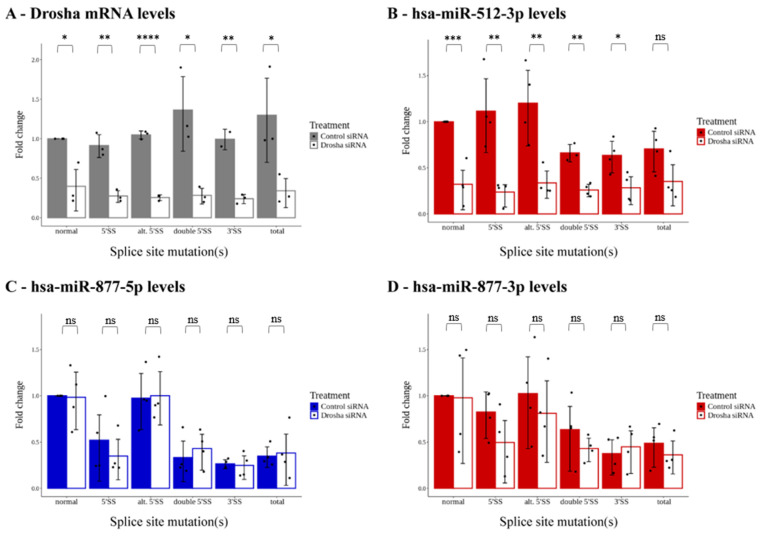
Investigating the role of Drosha on the expression of miRNAs from the different arms of the mirtron pre-miRNA. (**A**) Testing siRNA treatment by measuring Drosha mRNA levels. (**B**) Functional testing of Drosha knockdown by measuring the expression of the canonical miRNA, hsa-miR-512-3p. (**C**) Drosha knockdown did not affect the expression of the mirtron-derived hsa-miR-877-5p species. (**D**) Drosha knockdown did not have an effect on the mirtronic miR-877-3p either. Mean values of four experiments are shown on the graphs, the black dots indicate the results of individual experiments; error bars represent 95% confidence intervals. ns: not significant; *: *p* < 0.05; **: *p* < 0.01; ***: *p* < 0.001; ****: *p* < 0.0001.

## Data Availability

The original contributions presented in the study are included in the article, further inquiries can be directed to the corresponding author.

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
