# Peer review of "The Effect of Alternative Splicing Sites on Mirtron Formation and Arm Selection of Precursor microRNAs"

_ijms, 2024, doi:10.3390/ijms25147643_

Round 1

Reviewer 1 Report

Comments and Suggestions for Authors

The main question addressed by the research in this article is how alternative splicing mutations affect the formation and expression of miRNA species, with a particular focus on mirtrons and canonical miRNAs. The study investigates the interactions between the splicing apparatus and the miRNA maturation pathway. It aims to understand the effects of splicing site mutations on the processing and expression levels of miRNA species.

The alternative splicing mutations discussed in this study and their effects on miRNA formation are original and relevant to the field of molecular biology and RNA processing. The study focused on addressing a specific gap in scientific knowledge within the field. The presented research sheds light on the complex relationship between alternative splicing and miRNA biogenesis and makes important scientific contributions to little-known aspects of RNA editing.

The present research differs significantly from other published studies by focusing on the impact of alternative splicing mutations on mirtron formation and miRNA expression. Using EGFP reporters containing artificial introns, the study provides experimental evidence of how mutations in splice sites affect the processing of mirtrons compared to canonical miRNAs. This detailed analysis contributes to a better understanding of the mechanisms underlying mirtron maturation by providing new insight into the interplay between alternative splicing and miRNA biogenesis. The findings from the study enhance the existing knowledge base on RNA processing and provide an innovative perspective on the regulation of miRNA expression.

Using experimental data from EGFP-based reporter assays and quantitative real-time PCR analyses, the study provides evidence supporting the hypothesis that alternative splicing sites near major splice sites can affect mirtron processing and miRNA formation. On the other hand, the study reveals that mutations in splice sites can disrupt miRNA processing and the intricate relationship between alternative splicing and miRNA biogenesis, particularly in the context of mirtrons. In this context, the conclusions reached in the article align with the evidence and arguments presented in the study.

Overall, the results are supported by the experimental data and arguments presented in the study, providing insights into the regulatory mechanisms governing mirtron formation and miRNA expression in response to alternative splicing mutations. The references cited in the present article seem to align with the study's objectives and enhance the background knowledge and theoretical framework that support the research findings. The authors demonstrate a comprehensive view of the field by referring to previous studies and relevant literature. Considering all of these factors, it is appropriate to publish the article in its current form.

Author Response

Reviewer#1’s comments:

The main question addressed by the research in this article is how alternative splicing mutations affect the formation and expression of miRNA species, with a particular focus on mirtrons and canonical miRNAs. The study investigates the interactions between the splicing apparatus and the miRNA maturation pathway. It aims to understand the effects of splicing site mutations on the processing and expression levels of miRNA species.

The alternative splicing mutations discussed in this study and their effects on miRNA formation are original and relevant to the field of molecular biology and RNA processing. The study focused on addressing a specific gap in scientific knowledge within the field. The presented research sheds light on the complex relationship between alternative splicing and miRNA biogenesis and makes important scientific contributions to little-known aspects of RNA editing.

The present research differs significantly from other published studies by focusing on the impact of alternative splicing mutations on mirtron formation and miRNA expression. Using EGFP reporters containing artificial introns, the study provides experimental evidence of how mutations in splice sites affect the processing of mirtrons compared to canonical miRNAs. This detailed analysis contributes to a better understanding of the mechanisms underlying mirtron maturation by providing new insight into the interplay between alternative splicing and miRNA biogenesis. The findings from the study enhance the existing knowledge base on RNA processing and provide an innovative perspective on the regulation of miRNA expression.

Using experimental data from EGFP-based reporter assays and quantitative real-time PCR analyses, the study provides evidence supporting the hypothesis that alternative splicing sites near major splice sites can affect mirtron processing and miRNA formation. On the other hand, the study reveals that mutations in splice sites can disrupt miRNA processing and the intricate relationship between alternative splicing and miRNA biogenesis, particularly in the context of mirtrons. In this context, the conclusions reached in the article align with the evidence and arguments presented in the study.

Overall, the results are supported by the experimental data and arguments presented in the study, providing insights into the regulatory mechanisms governing mirtron formation and miRNA expression in response to alternative splicing mutations. The references cited in the present article seem to align with the study's objectives and enhance the background knowledge and theoretical framework that support the research findings. The authors demonstrate a comprehensive view of the field by referring to previous studies and relevant literature. Considering all of these factors, it is appropriate to publish the article in its current form.

Response:

We would like to thank Reviewer#1 for the very positive evaluation and for accepting the manuscript in its current form. However, based on the suggestions of other Reviewers, we have introduced some changes to the manuscript and hope that Reviewer#1 will still find it appropriate for publication in its revised form.

Reviewer 2 Report

Comments and Suggestions for Authors

Potential Points of improvement in this article

The maturation of miRNAs involves complex and varied pathways. While the canonical pathway is well-documented, alternative routes like Drosha- or Dicer-independent pathways add complexity. This complexity can lead to variations in miRNA maturation and function, making it challenging to predict miRNA behavior and their regulatory effects accurately .

The formation and function of miRNAs depend on multiple proteins and complexes, such as Drosha, DGCR8, and Dicer, among others. This dependence means that any mutations or malfunctions in these proteins can disrupt miRNA processing and function, potentially leading to diseases or developmental issues .

miRNA formation can be significantly affected by mutations, particularly in splice sites. Mutations in the splice sites of mirtrons, for example, can completely abolish miRNA formation, leading to a loss of gene regulatory functions and potentially contributing to disease states .

The exact mechanisms and structural details of mirtron processing are still not fully understood. This lack of complete understanding can hinder the development of targeted therapies or interventions that involve miRNAs .

The interaction between miRNA processing and the splicing machinery is intricate. Alternative splicing can produce different pre-miRNAs from the same locus, adding another layer of complexity to miRNA regulation. This can make it difficult to predict the outcomes of gene regulation mediated by miRNAs .

Due to the sequence complementarity requirement for target recognition, miRNAs can have off-target effects by binding to unintended mRNA targets. This off-target binding can lead to unintended gene silencing or dysregulation, complicating the therapeutic use of miRNAs .

The efficiency of miRNA-mediated gene silencing depends on the number of miRNA binding sites in the target mRNA's 3' UTR. This variability can result in inconsistent regulatory outcomes, which poses a challenge for designing miRNA-based therapeutic strategies .

The intricate regulatory network and the context-dependent nature of miRNA functions pose significant challenges for their therapeutic application. Delivering miRNAs to specific tissues or cells without affecting other cellular processes requires precise delivery mechanisms, which are currently difficult to achieve .

Alternative splicing can significantly impact miRNA processing, especially for mirtrons. The presence of alternative splice sites can hinder the production of functional miRNAs, which can complicate the regulatory landscape and potentially lead to dysregulation of gene expression .

The processing of mirtrons is strictly dependent on splicing, and any mutations in the flanking splice sites can render the mirtronic pre-miRNA inaccessible for Drosha cleavage. Structural constraints such as strong secondary structures or steric hindrance by associated RNA-binding proteins can further complicate miRNA processing .

The presence of a strong secondary structure in the pre-miRNA or the association of splicing factors can lead to the suppression of the upstream splice donor site. This interference can disrupt normal splicing patterns and potentially affect downstream processes.

Splice-site-overlapping pre-miRNA hairpins can engage in direct competition with the splicing machinery. This complexity in regulation can lead to misprocessing or inhibition of splicing, impacting the formation of mature miRNA molecules.

Some miRNAs, like Drosophila miR-1017, can recognize and bind to splice sites within their own introns, inhibiting splicing and preventing the formation of their own precursor molecules. This self-regulation can disrupt the splicing process and hinder miRNA maturation.

Despite advances in high-throughput studies, the structural features and regulatory mechanisms of mirtrons, particularly regarding splice site mutations and alternative splicing, remain incompletely understood. This lack of comprehensive knowledge hinders the accurate prediction of miRNA processing outcomes.

Mutations in major splice sites cannot always be compensated by nearby alternative splice sites, indicating a strict dependence on specific splicing machinery for mirtron processing. This limitation complicates efforts to manipulate splicing for therapeutic purposes.

Although genome-wide analyses suggest the prevalence of 5’-tailed mirtrons, their structural features and the responsible nucleases for removing the 5’ tail are not fully characterized. This gap in understanding complicates the design and manipulation of mirtrons for experimental or therapeutic purposes.

Mutations in splice sites can lead to effects on miRNA expression that are independent of Drosha activity. This suggests the involvement of additional regulatory mechanisms beyond canonical miRNA processing pathways.

Selective processing of 5p and 3p miRNA arms may be influenced differently by mutations in splice sites, leading to variable effects on miRNA expression levels. Understanding the factors influencing arm selection and stability is crucial for accurate interpretation of miRNA processing outcomes.

These points highlight the complexities and potential issues associated with miRNAs, despite their significant roles in gene regulation and potential therapeutic applications.

Comments on the Quality of English Language

Moderate editing of English language required

Author Response

Reviewer #2’s comments:

Potential Points of improvement in this article

The maturation of miRNAs involves complex and varied pathways. While the canonical pathway is well-documented, alternative routes like Drosha- or Dicer-independent pathways add complexity. This complexity can lead to variations in miRNA maturation and function, making it challenging to predict miRNA behavior and their regulatory effects accurately.

The formation and function of miRNAs depend on multiple proteins and complexes, such as Drosha, DGCR8, and Dicer, among others. This dependence means that any mutations or malfunctions in these proteins can disrupt miRNA processing and function, potentially leading to diseases or developmental issues.

miRNA formation can be significantly affected by mutations, particularly in splice sites. Mutations in the splice sites of mirtrons, for example, can completely abolish miRNA formation, leading to a loss of gene regulatory functions and potentially contributing to disease states.

The exact mechanisms and structural details of mirtron processing are still not fully understood. This lack of complete understanding can hinder the development of targeted therapies or interventions that involve miRNAs.

The interaction between miRNA processing and the splicing machinery is intricate. Alternative splicing can produce different pre-miRNAs from the same locus, adding another layer of complexity to miRNA regulation. This can make it difficult to predict the outcomes of gene regulation mediated by miRNAs.

Due to the sequence complementarity requirement for target recognition, miRNAs can have off-target effects by binding to unintended mRNA targets. This off-target binding can lead to unintended gene silencing or dysregulation, complicating the therapeutic use of miRNAs.

The efficiency of miRNA-mediated gene silencing depends on the number of miRNA binding sites in the target mRNA's 3' UTR. This variability can result in inconsistent regulatory outcomes, which poses a challenge for designing miRNA-based therapeutic strategies.

The intricate regulatory network and the context-dependent nature of miRNA functions pose significant challenges for their therapeutic application. Delivering miRNAs to specific tissues or cells without affecting other cellular processes requires precise delivery mechanisms, which are currently difficult to achieve.

Alternative splicing can significantly impact miRNA processing, especially for mirtrons. The presence of alternative splice sites can hinder the production of functional miRNAs, which can complicate the regulatory landscape and potentially lead to dysregulation of gene expression.

The processing of mirtrons is strictly dependent on splicing, and any mutations in the flanking splice sites can render the mirtronic pre-miRNA inaccessible for Drosha cleavage. Structural constraints such as strong secondary structures or steric hindrance by associated RNA-binding proteins can further complicate miRNA processing.

The presence of a strong secondary structure in the pre-miRNA or the association of splicing factors can lead to the suppression of the upstream splice donor site. This interference can disrupt normal splicing patterns and potentially affect downstream processes.

Splice-site-overlapping pre-miRNA hairpins can engage in direct competition with the splicing machinery. This complexity in regulation can lead to misprocessing or inhibition of splicing, impacting the formation of mature miRNA molecules.

Some miRNAs, like Drosophila miR-1017, can recognize and bind to splice sites within their own introns, inhibiting splicing and preventing the formation of their own precursor molecules. This self-regulation can disrupt the splicing process and hinder miRNA maturation.

Despite advances in high-throughput studies, the structural features and regulatory mechanisms of mirtrons, particularly regarding splice site mutations and alternative splicing, remain incompletely understood. This lack of comprehensive knowledge hinders the accurate prediction of miRNA processing outcomes.

Mutations in major splice sites cannot always be compensated by nearby alternative splice sites, indicating a strict dependence on specific splicing machinery for mirtron processing. This limitation complicates efforts to manipulate splicing for therapeutic purposes.

Although genome-wide analyses suggest the prevalence of 5’-tailed mirtrons, their structural features and the responsible nucleases for removing the 5’ tail are not fully characterized. This gap in understanding complicates the design and manipulation of mirtrons for experimental or therapeutic purposes.

Mutations in splice sites can lead to effects on miRNA expression that are independent of Drosha activity. This suggests the involvement of additional regulatory mechanisms beyond canonical miRNA processing pathways.

Selective processing of 5p and 3p miRNA arms may be influenced differently by mutations in splice sites, leading to variable effects on miRNA expression levels. Understanding the factors influencing arm selection and stability is crucial for accurate interpretation of miRNA processing outcomes.

These points highlight the complexities and potential issues associated with miRNAs, despite their significant roles in gene regulation and potential therapeutic applications.

Response:

We would like to thank Reviewer #2 for her/his comments and for highlighting the absence of the therapeutic aspect of mirtrons in the manuscript. We have included this aspect in the revised version based on the newly added publications #35 and #36 in the references, which are discussed in the lines 75-79 of the manuscript.

Comments on the Quality of English Language

Moderate editing of English language required

Response:

English language has been edited throughout the text as requested and the changes are indicated in the revised version of the manuscript.

Reviewer 3 Report

Comments and Suggestions for Authors

Manuscript ijms- ijms-3033034,

“The effect of alternative splicing sites on mirtron formation and arm selection of precursor microRNAs”.

 This study investigates the “non-canonical” mirtron pathway  of miRNA maturation, in particular the role of alternative splicing sites on mirtron formation. Currently, much remains unknown about miRNA formation and regulation of this process. This study provides new information to unravel the miRNA formation. Understanding miRNA formation and processing could be important for understanding immune system, neurological development, disease development and could be applied for drug development in the future. In general, this topic is of high importance and of considerable novelty.

 Overall, this manuscript represents and thorough and well-organized study at a high methodological level which was carefully presented in detail and clearly. Statistics and language are also of a high standard.

 There are some minor issues that need to be corrected. In my view, this manuscript can be accepted after minor corrections.

 Minor corrections:

1) More modern literature after 2020 should be included, described and discussed in the Introduction and Discussion sections.

2) I recommend to shorten the Discussion section or to include subsections in the Discussion. Now Discussion is too long and complex for understanding.

Also, I think that it is necessary to include a Conclusions part or at least a separate paragraph in Discussion with main conclusions.

3) EGFP and other gene names should be given in italics when you mean gene. For example, line 80, 82 for EGFP.

SREBF1 – line 96, NADi6 – line 104,

4) Please give full gene/protein name the first time you mention it, e.g. SREBF1, NADi6, POLR2A

5) correct site directed mutagenesis to site-directed mutagenesis on line 347

Author Response

Reviewer #3’s comments:

“The effect of alternative splicing sites on mirtron formation and arm selection of precursor microRNAs”.

This study investigates the “non-canonical” mirtron pathway of miRNA maturation, in particular the role of alternative splicing sites on mirtron formation. Currently, much remains unknown about miRNA formation and regulation of this process. This study provides new information to unravel the miRNA formation. Understanding miRNA formation and processing could be important for understanding immune system, neurological development, disease development and could be applied for drug development in the future. In general, this topic is of high importance and of considerable novelty.

Overall, this manuscript represents and thorough and well-organized study at a high methodological level which was carefully presented in detail and clearly. Statistics and language are also of a high standard.

There are some minor issues that need to be corrected. In my view, this manuscript can be accepted after minor corrections.

1) More modern literature after 2020 should be included, described and discussed in the Introduction and Discussion sections.

Response:

The literature on mirtrons after 2020 is rather limited, that is the reason why the orgininal reference list seemed less up-to-date. Nevertheless, we have collected additional publications from the period after 2020 and included them in the ‘Introduction’ and ‘Discussion’ sections, as requested. The newly discussed publications are listed in the updated reference section: #6-7, #12, #35-37, #47-48, #50-51. Related discussions are also integrated into the text in lines 43-44, 54-59, 75-82, 319-323, and 600-604 of the revised manuscript.

2) I recommend to shorten the Discussion section or to include subsections in the Discussion. Now Discussion is too long and complex for understanding.

Also, I think that it is necessary to include a Conclusions part or at least a separate paragraph in Discussion with main conclusions.

Response:

The discussion has been substantially re-written to make it shorter and more focused and comprehensive. In addition, a separate final paragraph is added to the end of the Discussion, serving as conclusion as requested.

3) EGFP and other gene names should be given in italics when you mean gene. For example, line 80, 82 for EGFP.

SREBF1 – line 96, NADi6 – line 104,

Response:

The full and shortened names of the genes and introns have been changed to italics, and the RNA and protein names are not italicized in the revised manuscript.

4) Please give full gene/protein name the first time you mention it, e.g. SREBF1, NADi6, POLR2A

Response:

We have provided the full name for DGCR8, EGFP, SREBF1, NDUSF1 and POLR2A at their first place of appearance in the manuscript as requested.

5) correct site directed mutagenesis to site-directed mutagenesis on line 347

Response:

The phrase is corrected as requested in line 363.

We thank Reviewer #3 for her/his valuable comments and suggestions and we hope that s/he will find the revised manuscript acceptable for publication.

Reviewer 4 Report

Comments and Suggestions for Authors

The authors investigated the splicing patterns of canonical miRNAs and mirtrons, a type of miRNA that cannot be processed by the Microprocessor complex, by expressing EGFR reporter constructs in human cells. It was found that mirtrons possess a different mature characteristic compared to the regular miRNAs. Overall, the results of this work are scientifically sound, confirming the uniqueness of mirtrons, and the discussion paves the road for future studies. I would recommend this manuscript to be published, if the following questions/issues could be addressed:

Line 112-114 reads confusing. Figure 1 shows that alternative 5’ SS mutations do not affect the translation of EGFR in all three constructs, and fluorescence is clearly shown.

Line 142-146 states that the mutations of all splice sites in miRNA-containing introns produce ‘stochastic background splicing noise’, which highly resembles the bands representing double 5’ SS splice products. Does it mean that they are also ‘stochastic background splicing noise’?

In Figure 3A, the difference between normal and double 5’ SS is reported to be significant, whereas that between normal and 3’ SS in 3B is nonsignificant, even though the pair clearly shows a higher fold change. Also, the difference between normal and 3’ SS mutant in 3D is obviously not nonsignificant. Please check and correct.

Similarly, the statistics in Figure 4A and B is not consistent within the panel. Please check again and correct.

Author Response

Reviewer #4’s comments:

The authors investigated the splicing patterns of canonical miRNAs and mirtrons, a type of miRNA that cannot be processed by the Microprocessor complex, by expressing EGFR reporter constructs in human cells. It was found that mirtrons possess a different mature characteristic compared to the regular miRNAs. Overall, the results of this work are scientifically sound, confirming the uniqueness of mirtrons, and the discussion paves the road for future studies. I would recommend this manuscript to be published, if the following questions/issues could be addressed:

Thank Reviewer#4 for her/his comments and for the overall positive evaluation..

Line 112-114 reads confusing. Figure 1 shows that alternative 5’ SS mutations do not affect the translation of EGFR in all three constructs, and fluorescence is clearly shown.

Response:

We agree with the comment that the original description of the mutation of the canonical and alternative 5’ SS is misleading in the manuscript.

If the canonical 5’ SS is unavailable because of a mutation, the splicing machinery efficiently uses the alternative 5’ SS, which is located 45 nucleotides upstream from the canonical 5’ SS. The usage of the alternative 5’ SS causes an in-frame, 45-nucleotide-long deletion in the exonic region of EGFP, leading to the loss of EGFP’s green fluorescence.

When the canonical 5’ SS is available and only the alternative 5’ SS is mutated by introducing a point mutation, the splicing machinery will use the canonical 5’ SS, and splicing continues to function without any disturbance. The point mutation of the alternative 5’ SS results in an in-frame, missense mutation that changes the amino acid from Valine to Leucine. This change in the amino acid sequence does not cause functional changes in EGFP fluorescence; therefore, we can see green fluorescence in the case of the mutation of the alternative 5’ SS of all three different intron types.

For better understanding, we clarified this issue in lines 143-146 and 174-179 of the revised manuscript.

Line 142-146 states that the mutations of all splice sites in miRNA-containing introns produce ‘stochastic background splicing noise’, which highly resembles the bands representing double 5’ SS splice products. Does it mean that they are also ‘stochastic background splicing noise’?

Response:

We attempted to Sanger sequence all of the splicing products, but unfortunately,  although they appeared as solid splicing products in the gels, they were still present in an amount unsuitable for sequencing and unambiguous identification.. It can be a result of stochastic splicing background or detection limitation of the methodology. The splicing products that we were able to sequence are shown in Supplementary Figure 1 and in the Snapgene files. Based on this, we believe it would be necessary to use more sensitive detection methods, such as deep sequencing of the splicing loci; however, this is beyond the scope of our current paper. To clarify this issue, we added a sentence highlighting our detection difficulties in lines 186-187 of the manuscript.

In Figure 3A, the difference between normal and double 5’ SS is reported to be significant, whereas that between normal and 3’ SS in 3B is nonsignificant, even though the pair clearly shows a higher fold change. Also, the difference between normal and 3’ SS mutant in 3D is obviously not nonsignificant. Please check and correct.

Response:

We employed rigorous statistical methods throughout our research, and the variations seen in Figure 3 A, B, and D are due to the standard deviation of the qPCR data, represented by error bars and the black points which represent the exact qPCR measurement data. When assessing the impact of splice site mutations on miRNA expression at specific loci, we utilized the one-sample t-test (Figure 3). This involved normalizing the miRNA expression of all mutants to the control, which was consistently set to 1, and comparing the fold change of the mutants against this baseline. This approach was essential due to the differences between the three biological replicates.

The fold-change observed between the normal and the double 5’ SS mutant in Figure 3A is statistically significant because the standard deviation of the ∆∆Ct values from the three qPCR measurements was moderate. However, in Figures 3B and 3D, the standard deviations were higher for the 3’ SS mutant compared to the normal condition, rendering the fold changes statistically insignificant despite their noticeable magnitude. We maintain that adhering to rigorous statistical approaches is crucial, even when they do not align with our initial hypotheses.

Similarly, the statistics in Figure 4A and B is not consistent within the panel. Please check again and correct.

Response:

To examine whether miRNA production from hsa-miR-877 mirtron loci in the case of splicing deficiency is dependent on Drosha, we tested the efficiency of Drosha knockdown (KD) on Drosha mRNA itself to prove that we successfully knocked down Drosha expression (Figure 4A). We normalized mRNA expression to the non-siRNA treated normal construct and pairwise compared the effect of Drosha KD for each mutation type using a two-sample t-test method. As shown in Figure 4A, Drosha siRNA treatment significantly lowered Drosha mRNA expression in every case of splice site mutations.

We applied the same comparison method when testing the effect of Drosha KD on miRNA production from the has-miR-512 canonical test miRNA to prove that without Drosha, the maturation of canonical miRNA is abolished from our expression constructs (Figure 4B). As shown in Figure 4B, Drosha KD significantly abolished miRNA production from our vectors except in the case of the total mutant.

The graphs represent the mean of four qPCR measurements, and the error bars represent standard deviation. If the standard deviation is greater, it will lower the significance. The number of * above the bars indicates the level of significance: *: p < 0.05; **: p < 0.01; ***: p < 0.001; ****: p < 0.0001. We believe that using the two-sample t-test to compare the mean values of two independent groups is statistically correct.

For better understanding, we added brackets to Figure 4 to show the pairwise comparisons between bars. The changes are shown in the new Figure 4 in the revised manuscript.

Round 2

Reviewer 2 Report

Comments and Suggestions for Authors

I'm sorry to let you know that your article has not been accepted for publication. While your work shows commendable effort and insight, it doesn't have the necessary depth and originality expected for this journal. The arguments presented lack clarity and fail to contribute significantly to the existing literature in the field. Also, the methodology does not support your conclusions, and the writing could benefit from more precise language and organization

Comments on the Quality of English Language

Moderate editing of English language required

Author Response

Responses to the comments and suggestions raised by the Reviewers

 Round 2

Reviewer #2:

Comments and Suggestions for Authors

I'm sorry to let you know that your article has not been accepted for publication. While your work shows commendable effort and insight, it doesn't have the necessary depth and originality expected for this journal. The arguments presented lack clarity and fail to contribute significantly to the existing literature in the field. Also, the methodology does not support your conclusions, and the writing could benefit from more precise language and organization

Comments on the Quality of English Language

Moderate editing of English language required

Response:

We are disappointed that in spite of the overall positive judgement during the first round of the revision, now Reviewer #2 does not feel that our manuscript is suitable for publication. As for the other Reviewers who were satisfied by the work and accepted the changes we made following their recommendations, we would also like to meet Reviewer #2’s expectations. Considering the new comments, we have carefully revised the manuscript, with special attention to improve the use of English language throughout the text. We corrected mistakes and re-phrased several sentences to make the presented findings and arguments more focused and more clear. We feel that such a careful perusal significantly improved our manuscript and we hope that the Reviewer can consider the revised version acceptable for publication.

Round 3

Reviewer 2 Report

Comments and Suggestions for Authors

I'm sorry to let you know that your article has not been accepted for publication. While your work shows commendable effort and insight, it doesn't have the necessary depth and originality expected for this journal. The arguments presented lack clarity and fail to contribute significantly to the existing literature in the field. Also, the methodology does not support your conclusions, and the writing could benefit from more precise language and organization.

Comments on the Quality of English Language

Moderate editing of English language required